# Falls from Height. Analysis of Predictors of Death in a Single-Center Retrospective Study

**DOI:** 10.3390/jcm9103175

**Published:** 2020-09-30

**Authors:** Alberto Casati, Stefano Granieri, Stefania Cimbanassi, Elisa Reitano, Osvaldo Chiara

**Affiliations:** 1General Surgery Unit, Policlinico Sant’Orsola-Malpighi, University of Bologna, Via Giuseppe Massarenti 9, 40138 Bologna, Italy; a.casati1988@gmail.com; 2General Surgery and Trauma Team, ASST Niguarda, Milano, Piazza Ospedale Maggiore 3, 20162 Milan, Italy; steff.granieri@gmail.com (S.G.); stefania.cimbanassi@ospedaleniguarda.it (S.C.); elisa.reitano@live.it (E.R.)

**Keywords:** trauma center, emergency medicine, emergency department, falls, mortality in falls

## Abstract

Falls from height (FFH) represent a distinct form of blunt trauma in urban areas. This study aimed to identify independent predictors of in-hospital mortality after accidental or intentional falls in different age groups. We conducted a retrospective study of all patients consecutively admitted after a fall in eight years, recording mechanism, intentionality, height of fall, age, site, classification of injuries, and outcome. We built multivariate regression models to identify independent predictors of mortality. A total of 948 patients with 82 deaths were observed. Among the accidental falls, mortality was 5.2%, whereas intentional jumpers showed a mortality of 20.4%. The death rate was higher for increasing heights, age >65, suicidal attempts, and injuries with AIS ≥3 (Abbreviated Injury Scale). Older patients reported a higher in-hospital mortality rate. Multivariate analysis identified height of fall, dynamic and severe head and chest injuries as independent predictors of mortality in the young adults’ group (18–65 years). For patients aged more than 65 years, the only risk factor independently related to death was severe head injuries. Our data demonstrate that in people older than 65, the height of fall may not represent a predictor of death.

## 1. Introduction

Falls from height (FFH) represent a distinct form of blunt trauma in which rapid vertical deceleration determines injuries to the victim’s body [1]. Falls account for almost 10% of all trauma admitted to the Emergency Department, mostly accidental (85%) [2] and in part intentional [3], the so-called “jumpers”. Outcomes and injury patterns depend on multiple environmental factors like height of fall, speed at impact, landing surface as well as victim’s characteristics such as age, position at impact, comorbidities, and intentionality of fall. As reported in the literature, suicidal attempts induce specific patterns of injuries [4] compared to accidental falls.

Various evidence available in the literature has already shown how increasing age and height of the fall correlate with worse outcomes [5,6]. The main aim of this study was to identify independent predictors of mortality in the study population and different age subgroups. As a secondary endpoint, the in-hospital mortality of patients of different age groups was evaluated.

## 2. Materials and Methods

The present study is a retrospective analysis of prospectively collected data of all trauma patients consecutively admitted after falling from height at ASST (Azienda Socio Sanitaria Territoriale) Niguarda Hospital, a level I Trauma Center in Milan, Italy, from October 2010 to December 2018. All details about trauma patients managed at Niguarda Hospital are collected in the Niguarda trauma registry, which is held by a Trauma Team consultant who is meant to keep it constantly updated, and is annually revised by the Head Department. The institution of trauma registry was approved by the Ethical Committee Milano Area 3 (record no. 534-102018). Given the retrospective nature of this study, specific board approval was not required.

Demographic data, Injury Severity Score (ISS), the height of fall, intentionality of fall, and survival outcome as stated by the Atlanta Trauma Registry Workshop guidelines [7] were retrieved from the registry. The height of fall was recorded in meters (m). The determination of height of the fall was obtained by directly asking patients and bystanders or was based on pre-hospital emergency medical staff (EMS) reports. Ground level falls were excluded from the analysis. Three different age groups were considered: pediatric age and teenagers (0–17 years), adults (18–64), and the elderly (≥65). We collected data on injuries of four anatomical sites according to the Abbreviated Injury Scale (AIS, 1998 version): head/face, chest, abdomen, and extremities. Critical head, chest, and abdominal defined by AIS ≥3 were identified. Injuries of extremities (long bones, pelvis) with AIS ≥ 2 were analyzed to also include the typical injuries related to low height falls. All cases with missing data or dead on the scene were excluded from further analysis.

Data were recorded in a computerized spreadsheet (Microsoft Excel 2016; Microsoft Corporation, Redmond, WA, USA) and analyzed with statistical software (IBM Corp. Released 2017. IBM SPSS Statistics for Windows, Version 25.0. Armonk, NY, USA, IBM Corp.). Categorical variables were explored with the χ^2^ test or Fisher’s exact test when appropriate. The distribution of continuous variables was assessed with normality tests and, since no variables had a normal distribution, differences among groups were evaluated with the independent sample Mann–Whitney test.

In order to identify independent predictors of mortality in different age groups, three multivariate regression models were built, providing adjusted odds ratios and 95% CI (Confidence Interval). All variables not contributing to the model (overfitting) were removed one-by-one from the model at each step based on the Nagelkerk R^2^ value. Multicollinearity was preventively assessed by examining the variance inflation factor (VIF). Models’ goodness of fit was explored with the Hosmer–Lemeshow test.

A two tailed *p*-value < 0.05 was considered statistically significant for all tests.

## 3. Results

Data of 948 patients were retrieved from the Trauma Registry. The median age was 42 years (Interquartile Range IQR 22–59). Six hundred eighty-two were males (71.9%) and 266 females (28.1%). The intentionality of the fall was obtained in all cases and was accidental in 733 cases (77.3%) and a suicide attempt in 215 cases (22.7%). The height of the fall was obtained or calculated for all patients with a median of 3 m (IQR 2–6). The median height was 2 m (IQR 2–4) for accidental falls (“fallers”) and 7 m (IQR 5–9) for intentional jumps (“jumpers”) (*p* < 0.001) (Table 1). The hospital mortality was 5.2% (38/733) in fallers and 20.5% (44/215) in jumpers (*p* < 0.001), with an overall death rate of 8.6%.

Table 1 summarizes the demographic and trauma-related characteristics of the study population. The comparison between fallers and jumpers showed a significatively greater proportion of females, ISS, chest AIS ≥ 3, abdomen AIS ≥ 3, and extremities AIS ≥ 2 injuries among jumpers as well as an increased median height of the fall.

As shown in Figure 1, the trend of deaths progressively rises for increasing clusters of heights.

Detailed results of comparisons between survivors and non-survivors are reported in Table 2.

A significantly higher proportion of deaths was recorded in the over 65 years group. Moreover, among non-survivors, we detected a greater height of the fall, an increased proportion of ASA 3–4 patients, a higher number of intentional jumpers, a higher rate of AIS ≥ 3 head, AIS ≥ 3 chest, AIS ≥ 3 abdomen, and AIS ≥ 2 extremity injuries. Similarly, we detected a significantly greater proportion of patients with ISS > 15 among non-survivors.

Variance inflation factor analysis pointed out significant multicollinearity (VIF > 2.5) between ISS and other variables meant to be entered in the multivariate models, therefore, it was excluded.

Height of the fall, ASA score, intentionality of the fall, head, chest, abdomen, and extremities injuries AIS ≥ 3 were selected to be entered in the regression models. Age was a further variable selected for the general population model.

Multivariate analysis failed to identify any independent predictors of death in the pediatric population. Intentionality, height of the fall, severe head and chest injuries resulted in independent predictors of death in the general and young adults’ group models. Furthermore, in the general population, age was identified as an independent predictor of mortality, with a 4% increase in the risk per year of age (*p* < 0.001; Odds Ratio—OR: 1.04; 95% CI: 1.03–1.06). Severe head injury was the only independent risk factor for mortality in the elderly group. Detailed results are reported in Table 3; Table 4.

All models were characterized by adequate goodness of fit. The overall predictive ability of all models was good, with remarkable results for the ones built for the general population (91.0%) and the young adults’ group (93.0%).

## 4. Discussion

This study showed in a large population that death after a fall can be independently predicted in the general population by age, the height of fall, intentionality, and severe chest and head injuries. In the elderly, the height of the fall cannot be considered as an independent predictor of death.

In Western countries, FFH represents one of the leading causes of admission to the Emergency Department [8], accounting for about 10% of all trauma patients [2]. This type of trauma represents the most common mechanism of self-inflicted injury [3], especially in the presence of specific underlying conditions such as psychiatric disease, drug, or alcohol abuse with a subsequent altered mental status. In urban areas, FFH represents the most frequent method of suicide attempt [9]. Conversely, in developing countries this event is generally accidental, with predominant involvement of workers in the setting of building constructions [10] and in pediatric patients during play activities in spring and summer seasons [2,4,9]. Specific biomechanical parameters in vertical deceleration trauma such as height of the fall, age of the victim, type of ground surface [11], and intentionality represent variables that lead to defined patterns of injuries and different outcomes [4,12,13]. Nevertheless, a precise reconstruction of the events and the gathering of accurate information can be challenging when eyewitnesses or clear signs on the scene are lacking. For these reasons, in our registry, only data reported by patients, bystanders, or pre-hospital emergency medical staff (EMS) were considered truly reliable.

In our study sample, we observed 82 deaths with an overall death rate of 8.6%. The study published by Goodacre et al. described a mortality rate of 1.4% [12], while our results were more similar to those reported by Liu C.C. et al., with a mortality rate of 22.7% [14] and Velmahos et al., who reported a mortality rate of 9.6% among fallers from more than 6 m [15].

In the univariate analysis, we detected significant differences in proportions/distribution for most of the variables taken into account. The height of the fall has been identified as a risk factor of unfavorable prognosis, analogously to other authors [8,10,15], Alizo et al. reported a 90% probability of death for falls exceeding 21 m (seven building stories) [16]. Our results were consistent with the literature with a median height of 6 m for non-survivors compared to 3 m for survived patients (*p* < 0.001). As already reported by other studies [3,4,17], and even in our analysis, a significantly different distribution of mortality between accidental fallers (38/733—5.2%) and intentional jumpers (44/215—20.4%) after vertical deceleration trauma (*p* < 0.001) has been noticed, both in adult and elderly age groups. In contrast to the higher rate of suicidal falls among elder women reported in previous work by our group [18], the present study pointed out a predominance of fallers among young adult males and a slight predominance of elder men among jumpers (*p* < 0.001). As shown in Table 1, we recorded a higher percentage of critical (AIS ≥ 3) chest and abdominal, and AIS ≥ 2 extremities injuries among jumpers (*p* < 0.001). Conversely, considering cephalic injuries, there were no differences between the two groups. Moreover, major trauma (ISS > 15) prevailed among the jumpers, confirming that greater heights are preferred by jumpers resulting in more severe injuries and worse survival outcomes.

We observed a linear growth of death rate by increasing height of fall due to higher energy dissipation at impact. The only exception was represented by elderly jumpers for whom our analysis failed to demonstrate this kind of proportional increase. Furthermore, looking at both young adults and elderly groups, we did not observe any lethal deceleration trauma among intentional jumpers for heights < 3 m (results not shown). As already reported by other authors [19], intentional jumpers from lower heights generally adopted a feet-first pattern of landing.

On the other hand, in case of accidental fall for the same height cluster, the orientation of the body during airtime cannot be controlled, leading to worse injuries. Indeed, as already reported by others [3,4,6,14], the first body region hitting the ground depends on body orientation during the time of flight, height, and intentionality of the fall. Suicidal attempts generally show as “feet-first” pattern of landing [4,15], whereas head injuries occur in unintentional falls with no control of the body during the fall. Our results show the unlikelihood of critical head injury occurring after low level jumps (Table 1), resulting in low mortality rates among jumpers compared to fallers below 6 m. We failed to demonstrate a higher predominance of lethal head injuries in the two groups due to the small number of deaths among jumpers.

As reported by Demetriades [5], mortality is considerably higher in elderly patients (>65 years old), probably due to increased rigidity of anatomical structures and consequent worse dissipation of kinetic energy. This result was corroborated by our survival analysis using Fisher’s exact test. These data confirm the idea that more aged patients are at increased risk of unfavorable prognosis [5], especially during the first hours, requiring the Trauma Team to keep high attention to evolving injuries while managing these frail patients.

To identify independent predictors of adverse survival outcome, we built four multivariate regression models. The general population model pointed out a 4% and 16% increase in the risk of death per year of age and meter of height, respectively. Suicidal attempts accounted for a nearly 3-fold increased risk of mortality. Patients sustaining severe head and chest injuries had a 7.6-fold and 2.7-fold increased risk of death, respectively. For the pediatric population, logistic regression analysis failed to detect any risk factors related to mortality. In the young adults’ subgroup, all aforementioned variables (but age) included in the general population model were confirmed as independent predictors of death. Of note, severe head injuries were related to nearly 9-fold risk of death. Our results demonstrated that patients aged more than 65 years sustaining severe head injuries are at risk of worse survival outcomes regardless of height of the fall and intentionality. This interesting finding, in contrast with those reported by some authors [5], demonstrates that age alone is a clear warning for the full activation of an experienced trauma team.

Nevertheless, to better interpret the apparently contradictory findings regarding the height of the fall in the general and >65 year-old models, one consideration should be done. For elderly patients, despite the relevant proportion of deaths, the average height of the fall was considerably lower compared to the adult population. This can explain why multiple regression failed to identify the height of the fall as an independent predictor of death in the elderly population.

Liu C.C. et al. identified only severe head injury AIS ≥ 4 as an independent predictor of mortality, whereas chest injuries did not seem to play any role in survival outcomes [14]. Interestingly, odds ratios per meter height decreased when age increased; this result can be explained by two considerations. The first age group only had four deaths (2% of the group), with a mean fall height of 3.55 m. Deaths among young adults numbered 43 (7.3% of the group), and the average height was 5 m. Finally, 35 patients died among the over 65-year-olds (21.3% of the group), with an average height of 3.48 m. Therefore, especially considering the young adults and elderly groups (which accounted for 95% of total deaths), it is reasonable to think that thee ORs decreased due to the drop in height of the fall.

To our knowledge, our study represents one of the largest single-center representations of vertical deceleration injuries at a level 1 trauma center in Europe with the review of the data of 948 patients collected in a standardized registry during an 8-year time interval. Moreover, our results seem to be comparable to those reported by other studies in the literature, especially considering the height of the fall and elderly age as independent predictors of mortality. Finally, our results were supported by a strict statistical methodology that led to the realization of multivariate models with a remarkable predictive ability.

This study has several limitations, first of all its retrospective nature. Furthermore, it did not take into account out-of-hospital mortality, which accounted for a significant proportion of deaths after FFH [17]. Furthermore, the limited size of the pediatric population and the even smaller number of deaths precluded any chance of identifying independent predictors of mortality in this age group. Finally, in suicidal attempts, it was extremely hard to determine psychiatric history or substance abuse before the fall.

## 5. Conclusions

The present study demonstrated that age, height of the fall, suicidal attempt, and severe head and chest injuries represent independent predictors of mortality. Considering different age clusters, we noticed that for elderly patients (over 65 years), the height of the fall, even though correlated with a greater proportion of deaths may not represent a predictor of mortality.

## Figures and Tables

**Figure 1 jcm-09-03175-f001:**
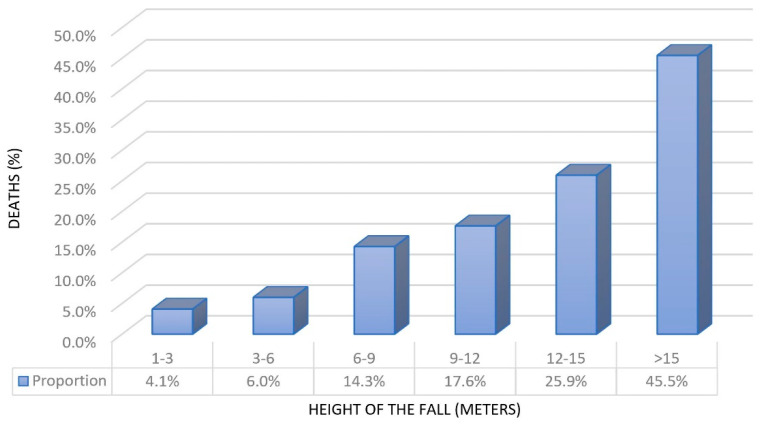
Trend of deaths per clusters of height.

**Table 1 jcm-09-03175-t001:** Demographic and trauma-related data.

Variables	Fallers (*n* = 733)		Jumpers (*n* = 215)		Total (*n* = 948)		*p*
	Value	%	Value	%	Value	%	
Gender							<0.001
Male	563	76.8	119	55.4	682	71.9	
Female	170	23.1	96	44.6	266	28.1	
Age (median—IQR)	42	19–60	39	26–55	42	22–59	0.732
Median age (IQR)							
Male	44	23–59	38	28–54	42	25–58	0.72
Female	33	11–63	41	22–57	38	17–61	0.161
ISS (median—IQR)	9	4–21	29	13–43	12	5–26	<0.001
Height of fall	2	2–4	7	5–9	3	2–6	<0.001
Anatomical districts of injuries (AIS ≥ 3)							
Head	183	25.0	66	30.7	249	26.3	0.095
Chest	179	24.4	128	59.5	307	32.4	<0.001
Abdomen	56	7.6	63	29.3	119	12.6	<0.001
Extremity (AIS ≥ 2)	278	37.9	166	77.2	444	46.8	<0.001
Outcome							<0.001
Survived	695	94.8	171	79.5	866	91.4	
Deceased	38	5.2	44	20.5	82	8.6	

IQR—Interquartile Range; ISS—Injury Severity Score.

**Table 2 jcm-09-03175-t002:** Comparisons between survivors and non-survivors.

Variables	Survived (*n* = 866)		Dead (*n* = 82)		Total (*n* = 948)		*p*
	Value	%	Value	%	Value	%	
Age (median—IQR)	40	20–57	60	40–76	42	22–59	<0.001
Age groups							<0.001
≤17	192	22.2	4	4.9	196	20.7	
18–64	545	62.9	43	52.4	588	62	
≥65	129	14.9	35	42.7	164	17.3	
Gender							0.125
Male	629	72.6	53	64.6	682	71.9	
Female	237	27.4	29	35.4	266	28.1	
ASA score							0.001
ASA ≤ 2	798	92.1	65	7.5	863	91.0	
ASA ≥ 3	68	7.9	17	20.7	85	9.0	
Intentionality of the fall							<0.001
Accidental	695	80.3	38	46.3	733	77.3	
Intentional	171	19.7	44	53.7	215	22.7	
Anatomical districts of injuries (AIS ≥ 3)							
Head	191	22.1	58	70.7	249	26.3	<0.001
Chest	245	28.3	62	75.6	307	32.4	<0.001
Abdomen	95	11.0	24	29.3	119	12.6	<0.001
Extremity (AIS ≥ 2)	389	44.9	55	67.1	444	46.8	<0.001
ISS (median—IQR)	9	4–22	42	29–59	12	5–26	<0.001
Height of the fall (m)—(Median/IQR)	3	2–5	6	3–9	3	2–6	<0.001

ASA—American Society of Anesthesiologists; AIS—Abbreviated Injury Scale; ISS—Injury Severity Score; TRISS—Trauma and Injury Severity Score.

**Table 3 jcm-09-03175-t003:** Multivariate analysis—general population.

Variables	*p*	OR	95% CI	
			Lower	Upper
Age	<0.001	1.04	1.03	1.06
Height of the fall (m)	<0.001	1.16	1.08	1.25
Intentional fall	0.003	2.73	1.42	5.26
Head injuries AIS ≥ 3	<0.001	7.64	4.28	13.64
Chest injuries AIS ≥ 3	0.003	2.69	1.41	5.13
Abdomen AIS ≥ 3	0.71	1.14	0.57	2.26
Extremities AIS ≥ 2	0.56	0.82	0.42	1.60
ASA score ≥ 3	0.26	1.52	0.73	3.13

AIS—Abbreviated Injury Scale; OR—Odds Ratio.

**Table 4 jcm-09-03175-t004:** Multivariate analysis—age groups.

**Age Group ≤ 17 (196 Patients)**				
**Variables**	***p***	**OR**	**95% CI**	
			**Lower**	**Upper**
Height of the fall (m)	0.164	1.607	0.824	3.134
Intentional fall	0.571	0.284	0.004	22.188
Head injuries AIS ≥ 3	0.994	-	-	-
Chest injuries AIS ≥ 3	0.992	-	-	-
Abdomen AIS ≥ 3	0.998	-	-	-
Extremities AIS ≥ 2	0.237	-	-	-
ASA score ≥ 3	0.999	-	-	-
**Age Group 18–65 (588 Patients)**				
**Variables**	***p***	**OR**	**95% CI**	
			**Lower**	**Upper**
Height of the fall (m)	0.002	1.12	1.05	1.20
Intentional fall	0.013	2.87	1.25	6.59
Head injuries AIS ≥ 3	<0.001	8.93	4.04	19.71
Chest injuries AIS ≥ 3	0.005	3.48	1.45	8.38
Abdomen AIS ≥ 3	0.83	1.10	0.46	2.61
Extremities AIS ≥ 2	0.78	0.88	0.36	2.15
ASA score ≥ 3	0.93	1.02	0.58	1.82
**Age group ≥ 65 (164 patients)**				
**Variables**	***p***	**OR**	**95% CI**	
			**Lower**	**Upper**
Height of the fall (m)	0.45	1.09	0.87	1.36
Intentional fall	0.067	3.24	0.92	11.41
Head injuries AIS ≥ 3	<0.001	5.45	2.10	14.17
Chest injuries AIS ≥ 3	0.19	1.93	0.72	5.17
Abdomen AIS ≥ 3	0.42	1.69	0.47	6.07
Extremities AIS ≥ 2	0.87	0.92	0.31	2.74
ASA score ≥ 3	0.13	1.64	0.86	3.12

AIS—Abbreviated Injury Scale; ASA—American Society of Anesthesiology.

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
