# Peer review of "Falls from Height. Analysis of Predictors of Death in a Single-Center Retrospective Study"

_jcm, 2020, doi:10.3390/jcm9103175_

Round 1

Reviewer 1 Report

The authors present an analysis of high fall trauma patients from a single center.

948 patients could be analyzed of which 77% were unintentional falls.

Not surprisingly, patients who jumped had a higher fall height, more injuries, and a higher mortality rate.

Univariate and multivariate analyses were provided for mortality. However, the subgroup approach of data analysis is limited here (see my comment regarding Table 2 below).

According to the methods section p-values for survival rates were derived from log-rank test (Kaplan-Meier curves). This is nice but not relevant here. K-M curves seek for difference in early or late deaths (like with cancer patients). Here in trauma patients only survival counts. Half of all trauma patients die within the first 24 hours in hospital, and the vast majority within the first week. It is thus not relevant whether a patients died after 2 or 4 days, but only if he or she finally survived. According to this, a simple Fisher Exact test would be more appropriate here.

Regarding the multivariate logistic models I strongly recommend some changes.

Due to the different sample size there is a different power in the three subanalyses which makes interpretation difficult. Conclusions like "Our data demonstrate that in people older than 65 the height of fall is not a predictor of death" in the text/abstract are definitely wrong. This is simply based on too few cases. Therefore, it is stronly recommended to present the regressions in the age subgroups.

Regarding the logistic model in all cases I recommend the following changes: 1. Include predictors known to be relevant for outcome from previous papers or scores, and present all potential predictors including the non-significant ones 2. age should not be used as a continuous predictor since it is well known that an age up to 55 years has no effect on mortality. 3. injury severity must be considered not only with two body regions; use either ISS or max AIS as predictor; head and thorax injury may still be added to see whether there is an additional effect due to these injuries. 4. ASA 3+ seems to be predictive but does not appear. 5. Height of fall and intentionality are the interesting factors here; even if they turn out to be not significant (in the new model), they should be presented. 6. Hosmer-Lemeshow p-value is nice but not too important here; it definitely will lower when sample size increases.

Minor points:

  • Fall from height (title) means that ground level falls should be excluded. Thus a height of 0m should not be considered. What was the lowest accepted height? I suggest to call the lowest category thus 1-3m.
  • Reference 5 was cited in the paper for the Atlanta Trauma Registry Workshop guidelines, but that reference contained high fall data from the NTDB
  • Table 1: if percentages were decided to be presented with one decimal, then please use 'xx.0' in order to separate it from rounded values
  • Table 2 is an attempt to show that age and height of fall is associated with mortality, separately for falls and jumps. It is a good example that with the given sample size such kind of analysis leads to very unsatisfactory results. There were too many subgroups, and too few deaths in many subgroups that the general effect could hardly be detected. It is strongly recommended to replace table 2 by a graph of height (m) versus mortality, in order to show that mortality increases with height of fall (i.e. no separation of type of fall and age)
  • Do not use more than 1 decimal for percentages (e.g., overall death rate 8.6%)

Author Response

The authors present an analysis of high fall trauma patients from a single center.

948 patients could be analyzed of which 77% were unintentional falls.

Not surprisingly, patients who jumped had a higher fall height, more injuries, and a higher mortality rate.

Univariate and multivariate analyses were provided for mortality. However, the subgroup approach of data analysis is limited here (see my comment regarding Table 2 below).

According to the methods section p-values for survival rates were derived from log-rank test (Kaplan-Meier curves). This is nice but not relevant here. K-M curves seek for difference in early or late deaths (like with cancer patients). Here in trauma patients only survival counts. Half of all trauma patients die within the first 24 hours in hospital, and the vast majority within the first week. It is thus not relevant whether a patients died after 2 or 4 days, but only if he or she finally survived. According to this, a simple Fisher Exact test would be more appropriate here.

Reply: we agree with the reviewer’s point of view. Time-to-event analysis is more appropriate when a long-term follow-up is intended, and the use of a survival method to analyze who dies first could be judged pointless. We modified our approach and replaced the Kaplan-Meier method with a Fisher’s exact test.

Regarding the multivariate logistic models I strongly recommend some changes.

Due to the different sample size there is a different power in the three subanalyses which makes interpretation difficult. Conclusions like "Our data demonstrate that in people older than 65 the height of fall is not a predictor of death" in the text/abstract are definitely wrong. This is simply based on too few cases. Therefore, it is stronly recommended to present the regressions in the age subgroups.

Reply: thank you for your comment. Actually, the subgroup of elderly patients accounts for 166 cases, which in our opinion cannot be defined “too few cases”.
We agree with the reviewer it is worth to report in tables 3 and 4 all the variables included in the multivariate model with the relative ORs and CIs. Nevertheless, it’s worth to underline that doing so, the stability of the model can be compromised (confirmed by an increase of the standard error from first to last block exceeding 10%). Moreover, the number of events (deaths) in the elderly group is formally too little to add 7 variables to the model.

Regarding the logistic model in all cases I recommend the following changes:

  1. Include predictors known to be relevant for outcome from previous papers or scores, and present all potential predictors including the non-significant ones

Reply: the current literature suggests age, height of the fall, circumstances of fall, fall orientation, landing surface/ elasticity and viscosity of tissue upon contact, head/neck AIS score 4. Unfortunately, in our study was not possible to include landing surface and force in the multivariate model due to the conspicuous proportion of missing data.

  1. age should not be used as a continuous predictor since it is well known that an age up to 55 years has no effect on mortality.

Reply: the reviewer is right; age has no effect on mortality up to 55 years. Nevertheless, clustering continuous variables in groups may result in a “statistical hack” that contributes to statistical power loss (Royston, et al., Dichotomizing continuous predictors in multiple regression: a bad idea, Statistics in Medicine, 25(1):127-141). Therefore, we decided to keep age as a continuous variable in multivariate analysis.

  1. injury severity must be considered not only with two body regions; use either ISS or max AIS as predictor; head and thorax injury may still be added to see whether there is an additional effect due to these injuries.
    4. ASA 3+ seems to be predictive but does not appear.

Reply: as specified in page 5, lines 121-122, Variance Inflation Factor analysis pointed out a potential confounding effect of ISS. Therefore, even though we are well aware of the importance of this variable in predicting survival outcome, it was excluded from regression models due to substantial multicollinearity with the other variables.
Abdomen and extremities injuries as well as ASA score were added to general and age-specific models as requested by the reviewer.

  1. Height of fall and intentionality are the interesting factors here; even if they turn out to be not significant (in the new model), they should be presented.

Reply: Height of fall and intentionality turned out to be independent predictors of death in the new model as well. Moreover in this new model Severe head injury was the only independent risk factor for mortality in the elderly group

  1. Hosmer-Lemeshow p-value is nice but not too important here; it definitely will lower when sample size increases.

If the reviewer deems it necessary, we can remove all the “models’ diagnostic” from table 4 (previously table 5)

Minor points:

Fall from height (title) means that ground level falls should be excluded. Thus a height of 0m should not be considered. What was the lowest accepted height? I suggest to call the lowest category thus 1-3m.

Reply: Actually, ground level falls were excluded from the analysis (we added this detail in page 2, line 70). The lowest height was 0.8 mt. We welcomed reviewer’s comment and modified the lowest category accordingly.

Reference 5 was cited in the paper for the Atlanta Trauma Registry Workshop guidelines, but that reference contained high fall data from the NTDB

Reply: we apologize for the mistake. The appropriate reference has been added in the reviewed manuscript as #23.

Table 1: if percentages were decided to be presented with one decimal, then please use 'xx.0' in order to separate it from rounded values

Reply: this has been changed

Table 2 is an attempt to show that age and height of fall is associated with mortality, separately for falls and jumps. It is a good example that with the given sample size such kind of analysis leads to very unsatisfactory results. There were too many subgroups, and too few deaths in many subgroups that the general effect could hardly be detected. It is strongly recommended to replace table 2 by a graph of height (m) versus mortality, in order to show that mortality increases with height of fall (i.e. no separation of type of fall and age)

We agree with the reviewer: table 2 may sound a little bit confusing. Therefore, we replaced it with a bar chart displaying the trend of deaths for increasing heights.

Do not use more than 1 decimal for percentages (e.g., overall death rate 8.6%)

Reply: this has been corrected.

Reviewer 2 Report

Thank you for the opportunity to review this interesting manuscript. Casati, et al., reviewed the trauma registry data for all patients who were admitted for falls from height over an eight year period. The stated aims of the study were to identify predictors of in-hospital mortality and the in-hospital mortality stratified by age groups. Their query of the registry produced 948 such patients with a hospital mortality of 8.6% overall, with 5.2% among unintentional falls and 20.4% among suicide attempts. Mortality was associated with increasing height of fall, increasing age, suicidal intention, and more severe injury. The authors conclude that in patients older than 65 years, height of fall is not a predictor of death.

The study seems reasonably executed. The writing and organization are very good. There are several points needing to be addressed:

First, what is the gap in the literature being addressed by this study? Height of fall and predictors of mortality are fairly well established in the literature. What does this paper add? This needs to be added to the introduction.

Why was a Kaplan-Meier analysis of death by hospital length of stay performed? What difference does it make if a patient dies on hospital day 3 or hospital day 7? This is not mentioned in the Introduction. It first appears in the fourth paragraph of the Methods section. The inclusion of a KM needs to be justified in the Introduction.

How was height of fall determined? This is an important aspect of the dataset and needs to be added to the Methods section.

The ISS was dichotomized and age was divided into three categories. In each case statistical power is lost. Important information is lost as well. For example, this method makes the age of 19 of equal mortality risk as 65, and a 3 year old at the same risk of death as a 17 year old given falls of similar height. In Table 1, how many 4 year olds attempt suicide by jumping from a height? See Royston, et al., Dichotomizing continuous predictors in multiple regression: a bad idea, Statistics in Medicine, 25(1):127-141

How many patients were excluded for missing information? Which variables were missing most often? Please comment on the bias introduced by excluding these patients.

Minor points:
1. The asterisk sumbol can be removed from the tables. The final sentence of the Methods states p<0.05 was considered significant. Alternatively, the significance can be entirely replaced by asterisk symbols. For example * = p<0.05, ** = p<0.01, *** = p<0.001. The readers do not need both.

2. The Odds Ratios and Confidence Intervals only need 2 digits following the decimal point. The additional precision does not improve the effect size.

Thank you once more for the opportunity to review this paper.

Author Response

Thank you for the opportunity to review this interesting manuscript. Casati, et al., reviewed the trauma registry data for all patients who were admitted for falls from height over an eight year period. The stated aims of the study were to identify predictors of in-hospital mortality and the in-hospital mortality stratified by age groups. Their query of the registry produced 948 such patients with a hospital mortality of 8.6% overall, with 5.2% among unintentional falls and 20.4% among suicide attempts. Mortality was associated with increasing height of fall, increasing age, suicidal intention, and more severe injury. The authors conclude that in patients older than 65 years, height of fall is not a predictor of death.

The study seems reasonably executed. The writing and organization are very good. There are several points needing to be addressed:

First, what is the gap in the literature being addressed by this study? Height of fall and predictors of mortality are fairly well established in the literature. What does this paper add? This needs to be added to the introduction.

Reply: The presented study is one of the largest European monocentric case series to our knowledge. Our data demonstrate that in people older than 65 years the height of the fall is not an independent predictor of death. This differs from the common belief that height of the fall is always a risk factor for mortality. Every emergency room doctor should be warned in case of a patient aged > 65 years sustaining fall trauma, irrespective of the height of the fall.

Why was a Kaplan-Meier analysis of death by hospital length of stay performed? What difference does it make if a patient dies on hospital day 3 or hospital day 7? This is not mentioned in the Introduction. It first appears in the fourth paragraph of the Methods section. The inclusion of a KM needs to be justified in the Introduction.

Reply: we agree with the reviewer’s point of view. Time-to-event analysis is more appropriate when a long-term follow-up is intended, and the use of a survival method to analyze who dies first could be judged pointless. We modified our approach and replaced the Kaplan-Meier method with a Fisher’s exact test.

How was height of fall determined? This is an important aspect of the dataset and needs to be added to the Methods section.

Reply: As explained in line 67-68 we considered 3 m the typical height of a normal building store. Medical staff on scene determines and report the expected height of the fall by asking bystanders or other information acquired when collecting patient’s anamnesis. If information could not be clearly obtained, the height of the fall was empirically estimated by observation of the scene (tree fall, presence of stairs …) (page 2, lines 68-69).
Line 166-167: “in our registry, only data reported by patients, bystanders or pre-hospital emergency medical staff (EMS) were considered truly reliable”

The ISS was dichotomized and age was divided into three categories. In each case statistical power is lost. Important information is lost as well. For example, this method makes the age of 19 of equal mortality risk as 65, and a 3 year old at the same risk of death as a 17 year old given falls of similar height. In Table 1, how many 4 year olds attempt suicide by jumping from a height? See Royston, et al., Dichotomizing continuous predictors in multiple regression: a bad idea, Statistics in Medicine, 25(1):127-141

Reply: The reason of dichotomization of ISS and representation of age in 3 clusters must be researched in the literature. Various manuscripts tend to categorize ISS due to the definition of major trauma (ISS ≥ 16). About age we decided to identify 3 groups according to other previous manuscripts published by our and many North American groups. However, we agree with the reviewer that grouping variables in clusters contribute to statistical power loss. Therefore, ISS and age have been reported as continuous variables in table 1, 2 (previously table 3, we replaced the old table 2 with a bar chart according to reviewer’s 1 suggestions) and multivariate analysis.

How many patients were excluded for missing information? Which variables were missing most often? Please comment on the bias introduced by excluding these patients.

Reply: All patients included in the study had the basic information extracted from the Trauma Registry about the dynamic of the fall (height and intentionality) and the survival outcome. Death on scene or deaths at arrival in ER were not included into the Trauma Registry. Therefore, we excluded a very few patients.

The most often missing information were physical parameters (patient’s body weight and height), landing surface (not reported or unspecific), hemodynamic parameters (Blood pressure, Hearth rate) and GCS. Anyway, none of the aforementioned variables were included in the comparative analysis. Therefore, the potential bias deriving by missing data is negligible.

Minor points:
1. The asterisk sumbol can be removed from the tables. The final sentence of the Methods states p<0.05 was considered significant. Alternatively, the significance can be entirely replaced by asterisk symbols. For example * = p<0.05, ** = p<0.01, *** = p<0.001. The readers do not need both.

Reply: this has been changed.

  1. The Odds Ratios and Confidence Intervals only need 2 digits following the decimal point. The additional precision does not improve the effect size.

Reply: this has been modified.

Round 2

Reviewer 1 Report

The authors presented a much improved revision of the paper.

I still would handle some details differently, but it is OK now.

For example, I still believe that age is better considered as a categorical variable. There is no general rule how to use continuous variables in a multivariate model. Height of fall is a good example here because there is an almost linear relation to mortality (see Figure). But if you include age as a continuous measurement you induce an imprecision in other places. But I do no longer insist on that.

The authors themselves realized the limitation of separate models in subgroups. Due to the limited number of deaths in the subgroups, the number of predictors in such sub-models decrease.

One interesting effect is that the odds ratio per meter height is decreasing when age increases.

I finally would like the authors to provige the average height of fall in the three age subgroups. And I assume that this height is lowest in the groupp with age>65.

Author Response

I still would handle some details differently, but it is OK now.

For example, I still believe that age is better considered as a categorical variable. There is no general rule how to use continuous variables in a multivariate model. Height of fall is a good example here because there is an almost linear relation to mortality (see Figure). But if you include age as a continuous measurement you induce an imprecision in other places. But I do no longer insist on that.

Reply: we agree with the reviewer about the possibility to add age as a continuous or a categorical variable. Eventually, we decided to keep age as a continuous variable according to reviewer’s #2 comments (listed below).

“The ISS was dichotomized, and age was divided into three categories. In each case, statistical power is lost. Important information is lost as well. For example, this method makes the age of 19 of equal mortality risk as 65, and a 3-year-old at the same risk of death as a 17-year-old given fall of similar height”

The authors themselves realized the limitation of separate models in subgroups. Due to the limited number of deaths in the subgroups, the number of predictors in such sub-models decrease.

One interesting effect is that the odds ratio per meter height is decreasing when age increases.

I finally would like the authors to provide the average height of fall in the three age subgroups. And I assume that this height is lowest in the group with age>65.

Reply: the reviewer perfectly detected the mathematics behind our results. The reason why odds ratios per meter height decrease when age grows can be explained with two considerations. The first age group has only 4 deaths (2% of the group), with a mean height of fall of 3.55 meters. Deaths among young adults are 43 (7.3% of the group) and the average height is 5 meters. Finally, 35 patients died among over 65-year-old (21.3% of the group), with an average height of 3.48 meters. Therefore, especially considering the young adults and elderly groups (which account for 95% of total deaths) it is reasonable to think that ORs decrease due to the drop in the height of the fall.

This consideration has been added to the manuscript (page 7, lines 207-213).

Reviewer 2 Report

Thank you for your consideration of my remarks and questions.

Author Response

Thank you for your consideration of my remarks and questions.

Reply: we are pleased our revisions have been appreciated.